# Characterization of the Tumor Microenvironment in Jaw Osteosarcomas, towards Prognostic Markers and New Therapeutic Targets

**DOI:** 10.3390/cancers15041004

**Published:** 2023-02-04

**Authors:** Hélios Bertin, Sophie Peries, Jérôme Amiaud, Nathalie Van Acker, Bastien Perrot, Corinne Bouvier, Sébastien Aubert, Béatrice Marie, Frédérique Larousserie, Gonzague De Pinieux, Vincent Crenn, Françoise Rédini, Anne Gomez-Brouchet

**Affiliations:** 1Department of Maxillofacial Surgery, Nantes University Hospital, 44000 Nantes, France; 2CRCI2NA-Nantes-Angers Cancer and Immunology Research Center, 44000 Nantes, France; 3Cancer Biobank of Toulouse, IUCT Oncopole, Toulouse University Hospital, 31100 Toulouse, France; 4UMR1246 SPHERE (MethodS in Patients-Centered Outcomes and Health ResEarch), Nantes University, 44000 Nantes, France; 5Department of Pathology, Timone Hospital, 13005 Marseille, France; 6Department of Pathology, Lille University Hospital, 59000 Lille, France; 7Department of Pathology, Nancy University Hospital, 54000 Nancy, France; 8Department of Pathology, Cochin Hospital, 75014 Paris, France; 9Department of Pathology, Tours University Hospital, 37000 Tours, France; 10Department of Orthopedic Surgery, Nantes University Hospital, 44000 Nantes, France

**Keywords:** osteosarcoma, mandible, maxilla, tumor microenvironment, tumor-associated macrophages, survival

## Abstract

**Simple Summary:**

Jaw osteosarcoma (JO) differs from its long-bones counterpart in many ways. The pathophysiology of this disease is still unknown, but the tumor microenvironment seems to play an important role in the progression of the disease and might offer new therapeutic perspectives. Through an immunohistochemical study performed on 50 biopsies of JO, we investigated various parameters associated with bone resorption, vascular and immune infiltrates. We demonstrated a strong and significant correlation between CD163 staining and lower survival in patients. Also, high levels of RANK and RANKL were found in the tumor samples and correlated with lower disease-free survival, while the T cells markers (CD4^+^ and CD8^+^) and the immune checkpoint PD-1/PD-L1 were poorly detected in the samples.

**Abstract:**

Background—The purpose of this study was to investigate the bone resorption, as well as the vascular and immune microenvironment, of jaw osteosarcomas (JO) and to correlate these features with patient clinical outcomes. Methods—We studied 50 JO biopsy samples by immunohistochemical analysis of tissue microarrays (TMAs). We investigated the bone remodeling markers RANK/RANKL/OPG, the endothelial glycoprotein CD146, and biomarkers of the immune environment (CD163 and CD68 of macrophages, CD4+ and CD8+ of tumor-infiltrating lymphocytes (TILs), and an immune checkpoint PD-1/PD-L1). The biomarkers were analyzed for their influence on progression (recurrence and metastasis), overall survival (OS), and disease-free survival (DFS). Results—A strong and significant correlation has been found between CD163 staining and lower OS and DFS. The level of CD4+ and CD8+ staining was low and non-significantly associated with survival outcomes. High levels of RANK and RANKL were found in the tumor samples and correlated with lower DFS. Conclusion—Our findings suggest that CD163+ TAMs represent markers of poor prognosis in JO. Targeting TAMs could represent a valuable therapeutic strategy in JO.

## 1. Introduction

Osteosarcoma is the most common type of malignant bone tumor. It preferentially affects the metaphysis of long bones during growth in children and adolescents [1]. Jaw osteosarcoma (JO) is rare, accounting for only 6%–13% of all osteosarcomas [1,2,3]. In the facial skeleton, osteosarcoma develops mainly in the mandible, and it differs from long-bone osteosarcomas (LBO) in many ways. First, it generally occurs two decades later than LBO, with a median onset of 35 years of age [4,5,6]. It also has a lower metastatic potential [1,3,7] and better overall survival (OS), reaching 77% at 5 years for patients with localized tumors and after complete carcinologic resection [2,7,8]. Finally, most JO are high-grade malignant lesions with a predominance of the chondroblastic histological subtype, while the osteoblastic form is more frequent in the limbs and the axial skeleton [2,5,6,9]. Due to the rarity of the disease, the treatment of JO is extrapolated from that of LBO [2,5]. The French standard multimodal treatment is based on neoadjuvant chemotherapy (neo-CT) combining high-dose methotrexate with etoposide-ifosfamide (M-EI) in children and adolescents, and doxorubicin-cisplatin-ifosfamide (API-AI) in adult patients [10,11]. However, in contrast to LBO, recent studies have not found a survival benefit for neo-CT in JO [12,13]. The adjuvant chemotherapy is adapted according to the histological response on the tumor resection, as assessed by the Huvos and Rosen score [14]. A good response is defined as a necrosis rate > 90% or by the presence of less than 10% viable tumor cells [15,16]. The use of external radiotherapy is not consensual, given the existence of radiation-induced forms of JO [17,18]. The standard of care remains carcinologic resection surgery with obtention of healthy margins, which represent the main prognostic factor for local control and survival [2,5,12,19]. Regardless of tumor removal and reconstruction, JO surgery leads to significant aesthetic and functional disabilities and affects the quality of life of the patients.

Osteosarcoma is a tumor of mesenchymal origin characterized by the proliferation of osteoblastic precursors and the production of osteoid matrix made of immature bone. The origin of primary osteosarcomas remains poorly understood. Osteosarcoma occurs in a complex, dynamic, and highly specialized bone environment made of osteoblasts, osteoclasts, and hematopoietic cells from which monocytes/macrophages derive within a mineralized extracellular matrix [20]. Crosstalk between tumor cells and the environment involves multiple signaling pathways, as evidenced by the existence of a vicious cycle involving the triad osteoprotegerin (OPG), receptor activator of NF-κB (RANK), and the ligand RANKL promoting tumor proliferation. Tumor cells secrete various factors, including RANKL, which binds to its receptor RANK on the surface of pre-osteoclasts, promoting their differentiation into mature osteoclasts. The thus activated osteoclasts degrade the bone tissue, allowing the release of factors present in the bone matrix, including TGF-β (transforming growth factor-β) and IGF (insulin-like growth factor), which are conducive to tumor cell proliferation [21,22]. OPG acts as a soluble decoy receptor for RANKL, preventing its binding to RANK and thus bone resorption. Tumor angiogenesis is mainly mediated by VEGF and is necessary for tumor growth and metastatic dissemination, especially to the lungs [23]. On the other hand, the formation of neo-vessels favors the penetration of neoadjuvant chemotherapy into the tumor niche [24,25]. The immune infiltrate of osteosarcoma is mainly composed of tumor-associated macrophages (TAMs) that regulate local immunity, angiogenesis, and tumor cell migration [26]. CD163^+^ TAMs are associated with better OS in patients with LBO [27]. On the other hand, the presence of CD163^+^ TAMs in the tumor infiltrate of head and neck squamous cell carcinoma (HNSCC) appears to be associated with metastasis and poor patient survival [28]. Tumor-infiltrating lymphocytes (TILs) represent the second most abundant cell type in the immune environment. CD8^+^ lymphocytes play a major role in delaying osteosarcoma metastases and can modulate the immune response of CD4^+^ helper [27]. The action of TILS can be inhibited by the tumor cells themselves or by activation of the PD-1/PD-L1 checkpoint leading to immune escape [29]. The bone microenvironment thus plays a prominent role in the development, progression, and chemoresistance of osteosarcoma [30,31]. While little is known regarding the specific environment of JO, the tumor microenvironment could, however, represent an interesting therapeutic target [32].

Through an immunohistochemical analysis of a large sample of tumors, the aim of this study was to investigate the microenvironment of JO regarding the bone resorption, as well as the vascular and immune parameters, to identify prognostic markers of the disease and potential therapeutic targets.

## 2. Materials and Methods

### 2.1. Patient and Tumor Characteristics

The samples were derived from human biopsies of JO and collected as part of a collaborative project between different French centers that are members of the “Groupe Sarcome Français-Groupe d’Etude des Tumeurs Osseuses“ (GSF-GETO), the Rare Cancer Network (RCN), and the “Réseau d’Expertise Français des Cancers ORL Rares“ (REFCOR). Written consent was obtained from each patient for the collection of their biological samples, or from their guardian in case of patients under 18 years of age, in compliance with the bioethics laws and the declaration of Helsinki. In keeping with French legislation, the biobank cancer collection was declared to the Ministry of High Education and Research (DC-2008-463 and DC-2020-474), and a transfer agreement was obtained (AC-2020-4031, last approval 06/01/2021) after approval by the relevant ethics committees. Tissue microarrays (TMA) were prepared from diagnostic biopsies of 50 patients and stored at the certified NF 96-900 cancer biobank of Toulouse (BB-0033-00014). For each biopsy, a triplicate sampling of 1 mm diameter was taken from the tumors in the areas with the highest number of tumor cells and placed in a new block with a Tissue Arrayer MiniCore^®^ (Excilone, Elancourt, France). The block was then cut into 4 μm sections with a conventional microtome. All patient records and data were anonymized. The patient data included age, gender, date of diagnosis, histologic subtype, and therapeutic response evaluated on tumor resection. We also collected data on local or metastatic recurrence, date of death, and survival.

### 2.2. Immunohistochemistry

Immunostaining was performed with antibodies directed against RANK, RANKL, OPG, CD146, CD68, CD163, CD4^+^, CD8^+^, and PD-1 using a Discovery Ultra automated research platform (Ventana Medical Systems, Arizona, USA) and against PD-L1 on an Autostainer Link 48 from Dako (Agilent USA, Denmark). The steaming and deparaffinization steps programmed into the Discovery Ultra device consisted of heating the slides to 60 °C for 8 min, followed by the application of a ready-for-use Tris-acid solution (EZ Prep solution, Ventana) (three washes for 8 min) at 69 °C. For CD68 staining, sections were pre-treated with protease 1 (Ventana) for 4 min at 37 °C. For OPG and RANK staining, slides were pretreated in a pH = 6 citrate buffer (Ventana) for 20 min. For the other markers (RANKL, CD146, CD163, CD8^+^, and PD-1), sections were pretreated with Tris-EDTA (pH = 8–8.5, Ventana) for 20, 20, 64, 32, and 64 min, respectively. Endogenous peroxidase activity was blocked using CM (ChromoMap) inhibitor for 32 min at 37 °C (Ventana). The samples were then incubated with primary antibodies against CD146 (EPR3208), OPG, RANK (H-300), RANKL (N-19), CD68 (PG-M1), CD163 (MRQ-26), CD8^+^ (clone SP57), and PD-1 (NAT105) at 36 °C for 60 min for CD146/OPG/RANK/RANKL, or 20 min, 32 min, 20 min, and 16 min for the other markers, respectively. Staining was performed with a Ventana kit (secondary antibody conjugated to streptavidin–peroxidase) for 16–60 min at 37 °C. Sections were stained by incubation in a diaminobenzidine and H_2_O_2_ solution for 7 min at room temperature. The slides were then stained with hematoxylin (Ventana) and finally rinsed with water, dehydrated (ethanol and xylene), and then mounted.

For PD-L1 staining, sections were dried for 1 h at 58 °C followed by overnight at 37 °C. The sections were then deparaffinized with toluene and rehydrated in ethanol, followed by pretreatment with a high pH target retrieval solution (DAKO, EnVision Flex, Denmark), and a heat-based antigen retrieval method was used before incubation. Endogenous peroxidase activity was blocked by incubation for 5 min in 3% H_2_O_2_. The primary antibody was used at a 1:500 dilution (CliniSciences, Nanterre, France; anti-PD-L1 clone E1L3N) for 20 min at 37 °C. Stainings were performed with an Envision kit (DAKO, Carpinteria, CA, USA) by incubation in a diaminobenzidine solution for 10 min followed by staining with hematoxylin for 5 min.

Immunoreactivity was considered positive if detected in > 1% of the cells per core of 1 mm, irrespective of the staining intensity. As previously described [27], for each marker a percentage of stained cells was defined, except for the CD146 marker of angiogenesis, which was classified based on either more or less than 50% of the cells stained. For the CD163 marker of TAMs, CD68 of osteoclastic cells (pre-osteoclastic small cells and giant cells/mature osteoclasts), and CD146 of endothelial cells, the staining was considered “high” when ≥ 50% of the cells were positive per core. A threshold of 10% was used for the markers RANKL and CD4; 1% for CD8 lymphocytes, PD-1, and PD-L1. Positive controls were assessed on lymphoid nodes for the CD8, PD-1, and PD-L1 antibodies, on renal samples for the CD146 antibody, and on giant cell tumors of bone for the other markers. A double-blind examination by two pathologists who are experts in bone sarcoma was carried out.

### 2.3. Statistical Analysis

The data are summarized as the frequency and percentage for the categorical variables and the median and range for the continuous variables. Correlations between the quantitative data were assessed using Spearman’s rank correlation coefficient. Links with the diagnostic status or the histological response were assessed with Fisher’s test for the categorical covariates and the Mann–Whitney U test for the quantitative covariates. OS was defined as the time from inclusion to death from any cause (event) or the last follow-up (censored data). Disease-free survival (DFS) was defined as the time from inclusion to metastatic progression or death (event) or the last follow-up (censored data). All survival rates were estimated by the Kaplan–Meier method with 95% confidence intervals (CI). Two-sided *p*-values < 0.05 were considered statistically significant. The statistical analysis was performed using GraphPad Prism 9.0 software for Mac (GraphPad Software, La Jolla, CA, USA).

## 3. Results

### 3.1. Patient Characteristics

Of the 50 patients included, the mean age at diagnosis was 47.8 ± 19.9 years of age, and the sex distribution was 28 males versus 22 females. Most of the tumors were high-grade osteosarcomas of the maxillary or the mandibular bone, with a strong predominance of the chondroblastic subtype. In three patients, the osteosarcoma was secondary to irradiation (two cases) and to a rhabdomyosarcoma (one patient), all of the other tumors were primary lesions. A neo-CT was administered in 26 patients, including the M-EI protocol in three patients aged 17.3, 22, and 22.4 years and the API-AI protocol in 18 other patients aged 28.7 to 72.5 years, while the treatment was unknown in 5 patients secondarily excluded from the immunohistochemical analysis. Twenty patients were considered to be poor responders to the neoadjuvant chemotherapy. The surgical resection margins were known for 36 patients: 17 were classified as R0 (i.e., without tumor remnant), 14 were scored as R1 (microscopic invasion of the margins), 5 were scored as R2 (macroscopic invasion), while for 14 patients these data were missing, or the patients had not undergone surgical resection. Local relapse occurred in 6 patients (12%), with a mean delay of 556.3 ± 439.6 days. Of these patients, 4/6 were classified as R1, 2/6 as R0, and 4/6 were poor responders to neoadjuvant chemotherapy. Lung metastases occurred in 7 patients (14%), after a mean of 625.2 ± 626.2 days (one patient presented metastasis at diagnosis). The mean OS and DFS reached 1600 ± 1737 days and 1469 ± 1745 days, respectively. Fourteen patients were dead at the end of the follow-up period. All of the epidemiological data are presented in Table 1.

### 3.2. Immunohistochemical Analysis

Eleven samples were eliminated from the analysis because of severely degraded material. Regarding the bone remodeling markers, the staining was intense for the RANK and RANKL biomarkers, with staining greater than 10% per core in 22/27 and 30/35 samples, respectively (Table 2). The OPG staining was not interpretable due to non-specific background signal. For the staining of blood vessels, we observed that 20/35 samples exhibited CD146 staining higher than 50%. With regard to the immune environment, 9/28 and 3/28 patients exhibited staining greater than 50% per core for the CD163 and CD68 markers, respectively. The CD4+ and CD8+ staining was low in all samples analyzed, with a mean number of positive cells of 13.7 ± 19.8 and 8.9 ± 15.2, respectively. The PD-1 and PD-L1 stainings yielded comparable results, with no staining in more than 95% of cases (Figure 1).

Correlations between the biomarker stainings are presented in Table 2. The immune biomarkers were correlated together. CD163 and CD68 were highly correlated (r = 0.69, *p* < 0.0001), as were the CD163 and CD8+ markers (r = 0.67, *p* = 0.0003). Moreover, correlations were found between CD163 and CD4+, CD4+ and CD68, and RANK with CD4+.

### 3.3. Biomarkers and Clinical Parameters Associated with the Diagnosis and Histological Response

The biomarkers tested were not associated with the histologic subtype or tumor grade (high or intermediate). Regarding the response to the neoadjuvant chemotherapy, there was no correlation between the biomarkers and the status “good” or “poor” responder. We observed a non-significant association between a high level of RANKL (≥ 10%) and a poor response to the neoadjuvant chemotherapy (OR = 6.50, 95% CI [0.57–99.74], *p* = 0.20).

### 3.4. Clinical Parameters and Biomarkers Associated with Overall Survival

After a mean follow-up of 32.9 months, 14 patients (35.9%) had died. The 5-year OS rate was estimated to be 52.4% (95% CI [30.0–70.7]). Univariate analysis showed that a poor response to the chemotherapy was associated with an unfavorable OS (*p* = 0.02) (Table 3). A high level of CD163-positive cells (≥ 50%) in biopsies was significantly correlated with a lower OS in patients (*p* = 0.0006). A trend of worse survival was also observed for patients with ≥ 50% CD68- and ≥ 10% of CD4+-positive cells (p1 = 0.04 and p2 = 0.002) (Figure 2).

### 3.5. Clinical Parameters and Biomarkers Associated with Disease-Free Survival

Post-treatment events (local progression and/or metastases) occurred in 9 patients (23%). The 5-year DFS rate was estimated to be 59.8% (95% CI [33.7–78.5]). Univariate analysis showed that a high level of RANK and RANKL (≥ 10%) correlated with a lower DFS (p_1_ = 0.03 and p_2_ = 0.03, respectively). A trend of worse survival was also observed for patients with ≥ 50% CD163- and CD68-positive cells (p_1_ = 0.003 and p_2_ = 0.04, respectively) (Figure 2).

## 4. Discussion

JO is a specific disease with distinct differences from LBO. As the survival of osteosarcoma patients has not changed significantly in the past 30 years [33], further investigation is needed to search for prognostic markers of the disease and potential new therapeutic targets. In the absence of well-established mechanisms of oncogenesis, the tumor microenvironment represents a target of choice [26]. The use of patient biopsies is necessary because of the tumor heterogeneity, the rarity of the disease, and the need to model the interactions between the tumor and its surroundings. Our immunohistochemical analysis included the largest series of JO published to date. The mean age of the patients in our series is comparable to that in the literature, which reports onset at approximately 33–46 years of age [1,3,5,19]. The predominance of the chondroblastic subtype is also in agreement with previously published data [2,12,34,35]. The neo-CT was administered according to the French standard treatment protocols, the M-EI chemotherapy being reserved for the youngest patients (i.e., < 25 years old), while the others receive the API-AI regimen. We assume that for a large proportion of the patients for whom neo-CT data were not available, they underwent primary tumor removal surgery, as suggested by recent publications that did not find a consistent survival benefit for neo-CT over primary surgery in the management of osteosarcomas of the head and neck [12,13]. Neo-CT could also delay surgical treatment, leading to non-operable patients. The low number of good responders (12%) in our series supports the lower efficacy of neo-CT in JO compared to the 61.1% of good response observed in long bone localizations [27]. The margins of resection were affected for 19 of the 36 patients for whom this information was available. This underlines the difficulty in obtaining clear margins in craniofacial locations. While the commonly accepted bone margin on JO is 20 mm on the bone, it is reduced to 2 mm on the soft tissue [2,16]. The OS of patients was less than that commonly seen in LBO, while the DFS was slightly better than in LBO [27].

Our results have identified a strong and significant correlation between CD163 staining and a lower OS in JO, and the same trend was found regarding the CD68 biomarker. CD163 is a glycoprotein receptor present on the surface of monocyte-macrophages including the M2 subtype. To better address the M2 response, some authors favour the use of the CD163^+^-Erythropoietin Receptor (EPOR^+^) co-staining, as identified in human osteosarcoma lung metastasis, or the CD163/CD11c ratio [36,37]. CD68 is a pan-macrophage marker capable of detecting macrophages of any subtype as well as osteoclasts. Our results are in accordance with studies performed in soft tissue tumors such as HNSCC, in which CD163^+^ TAMs correlate with poorer patient survival [28]. Shiraishi et al. correlated the high percentage of CD163^+^ cells with decreased OS and a higher histologic grade in 62 pleiomorphic undifferentiated sarcoma samples [38]. However, with regard to osteosarcomas that occur within hard and mineralized tissue, the role played by macrophages is still relatively unknown and the results remain contradictory [39]. The immunohistochemical study by Gomez-Brouchet et al. performed under the same technical and methodological conditions as ours showed a significant correlation between the high (> 50%) level of CD163-positive cells in biopsies and a higher overall survival (*p* = 0.0025) in 124 samples from LBO [27]. Several hypotheses can be advanced to explain the differences in the microenvironment between JO and LBO. The first lies in different ossification mechanisms and embryological origin, as the facial bones are derived from neuro-ectodermal tissue while the axial skeleton has a mesodermal origin [32,40]. Moreover, bone remodeling and regeneration are faster in craniofacial localization [41,42]. The differences in biomechanical properties between the mandible and the long bones could also explain differences in tumor behavior, as osteosarcomas have the particularity of occurring within a solid tissue, with the lesion being more compliant than the tissue in which it originates. The loss of tissue architecture during malignant progression causes alterations in the mechanical stimuli to which osteosarcoma cells are exposed [43]. Furthermore, it has been demonstrated that mechanical forces are able to inhibit the vascular endothelial growth and bone growth at the end of the adolescence, thus suggesting new therapeutic options for diseases with aberrant activity of bone and vessels such as osteosarcoma [44]. Regarding the particular cell subtype of TAMs, it has been demonstrated the existence within the tumor of bipotent CD68^+^ and CD163^+^ cells capable of differentiating into both osteoclasts and macrophages with different effects on tumor progression and immunodepression [45]. The result concerning CD163^+^ TAMs is of clinical interest as it represents a prognostic marker in JO, and it could also represent a potential target for immunotherapies. The macrophage-activating agent muramyl tripeptide-phosphatidylethanolamine (MTP-PE) or mifamurtide represents an adjuvant treatment to chemotherapy in osteosarcoma. However, its use in metastatic patients does not appear to be effective in increasing survival [46,47]. As previously described by Alves et al. in an immunohistochemical analysis of 21 samples of JO, the level of CD4^+^ and CD8^+^ staining across the patient samples was low [48]. A non-statistical association was found between CD4^+^ infiltration (only 8 samples expressing ≥ 10% of cells stained) and lower OS and DFS in patients. CD4^+^ cells are instead associated with better survival in patients with LBO, highlighting their protective effect in regard to tumor progression [49]. CD8^+^ TILs are generally thought to have a significant impact on patient survival in osteosarcoma [27,48,49]. As in the study by Alves et al., we did not find an association between CD8^+^ staining and clinical parameters in JO [48]. Finally, immune checkpoint PD-1/PD-L1 staining was almost absent, and this result is consistent with the lack of a significant effect on survival obtained with PD-1/PD-L1 inhibitors [50].

Our other results concern the bone remodeling biomarkers and vascularization. High levels of RANK and RANKL were found in the tumor samples and correlated with a lower OS and DFS. These results are consistent with the expression of these markers in many solid tumors, and their association with tumor cell migration and metastatic dissemination [51]. In a recent phase II study investigating the use of denosumab in giant cells tumors of bone (GCTB), Palmerini et al. highlighted that elevated baseline s-CTX, a bone resorption marker, was associated with a higher risk of progression of the disease [52]. In vivo studies have shown that blocking RANKL by using truncated OPG or by gene inactivation in genetically engineered mice slows tumor growth and improves animal survival [53,54]. Although denosumab has been shown to have an anti-invasive effect in vitro on osteosarcoma cell lines (U2OS, MG-63), it has not been tested in humans because of a possible reduction in the efficacy of chemotherapy and the deleterious effects observed with zoledronic acid in LBO [55,56]. CD146 represents a biomarker of tumor angiogenesis, and its expression is directly correlated with tumor progression, invasion, and metastasis in osteosarcoma [57]. Although there was strong CD146 staining in our series, we did not find evidence of a correlation with patient survival or clinical parameters. A previous immunohistochemical study targeting VEGF showed a lower level of vascularisation in JO than in LBO, which may explain the lower metastatic potential of the former [34]. The role of angiogenesis in tumorigenesis is not only related to dissemination propensity. It also appears to be correlated with the bone resorption, as evidenced in GCTB, with involvement of VEGFR in supporting RANKL-induced osteoclastogenesis [58,59]. In this regard, there is recent evidence regarding the greater efficacy of combination of the anti-VEGFR lenvatinib and the anti-RANKL denosumab in the treatment of GCTB primary cultures, compared to denosumab alone [60]. Nevertheless, if this therapeutic combination has shown its effectiveness on GCTB osteolytic lesions, its efficacy has not yet been demonstrated on osteosarcomas, which are much more osteogenic tumors, and verified in patients. Moreover, the potential efficacy of tyrosine kinase inhibitor treatment is also emerging from recent studies in refractory, relapsed, or metastatic osteosarcoma [61,62]. Lenvatinib and pembrolizumab are also being studied in ongoing trials of advanced sarcoma (NCT04784247).

This immunohistochemical study suffers from some limitations. The first is the small number of JO samples, which is due to the rarity of the disease, although it remains the largest series to date. The second limitation is due to missing data because of technical considerations or due to patients lost to follow-up as a result of the retrospective nature of the study. The third concerns the selection of tumor areas during the biopsy and the construction of the TMAs, which may not reflect the tumor niche in its entirety and complexity. The biopsy is performed by a specialized team in a reference center [63], taking into account the accessibility of the lesion, the ability to secondarily remove the biopsy tract and to avoid mineralized bone areas rather than select the bone-tumor interface which is of interest for the study of the microenvironment. Nevertheless, for JO the tumor biopsy is often facilitated by the presence of endobuccal swelling with an easy-to-access bone-tumor interface under the oral mucosa. Finally, the heterogeneity of the fixation and decalcification techniques could potentially lead to different antigenic expressions. However, one of the main advantages lies in the use of TMAs, as this standardizes the immunohistochemical assay and ensures comparability of the samples for the biomarkers studied.

## 5. Conclusions

Our findings suggest that CD163^+^ TAMs represent markers of poor prognosis in JO. Systematic analysis of CD163 expression performed on JO biopsies at diagnosis may hence allow stratification of patients and monitoring of the response to therapy. Targeting TAMs may also be a valuable therapeutic strategy in JO. RANK and RANKL pathways appear to be involved in disease progression and metastatic dissemination in JO, and thus constitute a novel potential therapeutic approach.

## Figures and Tables

**Figure 1 cancers-15-01004-f001:**
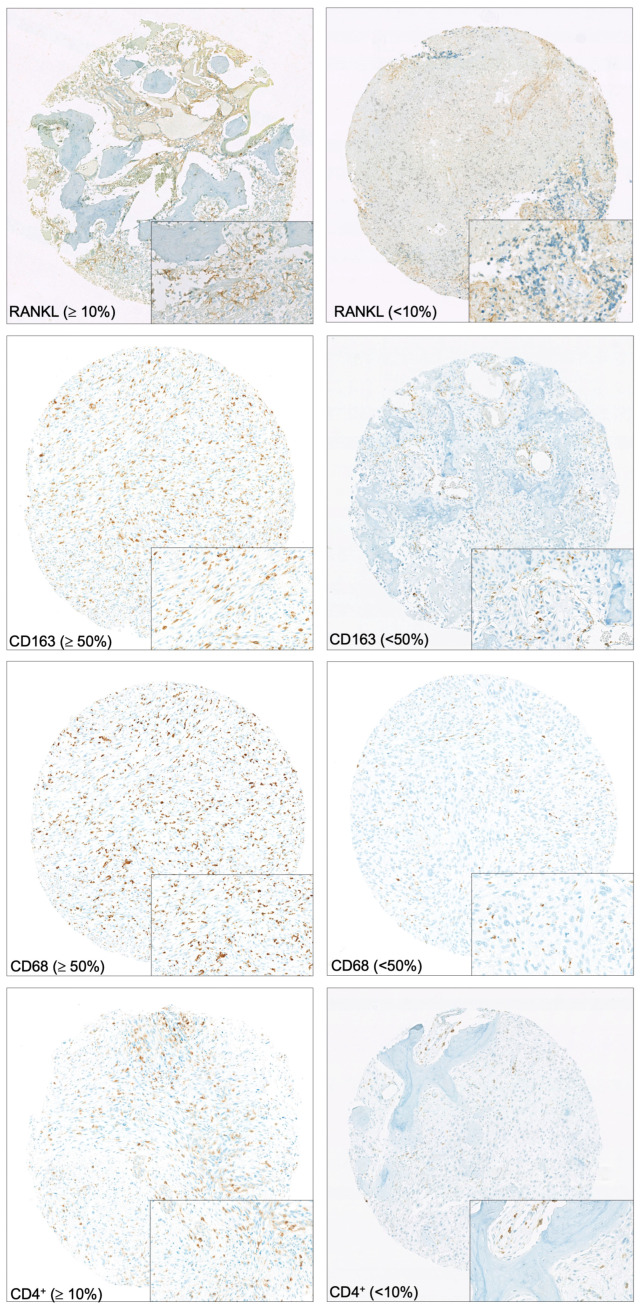
Immunohistochemical staining. Sample images of tissue microarrays prepared from patient biopsies and stained for RANKL, CD163, CD68, and CD4^+^ (magnification ×7). Frames correspond to a high-power field of each image (magnification ×40).

**Figure 2 cancers-15-01004-f002:**
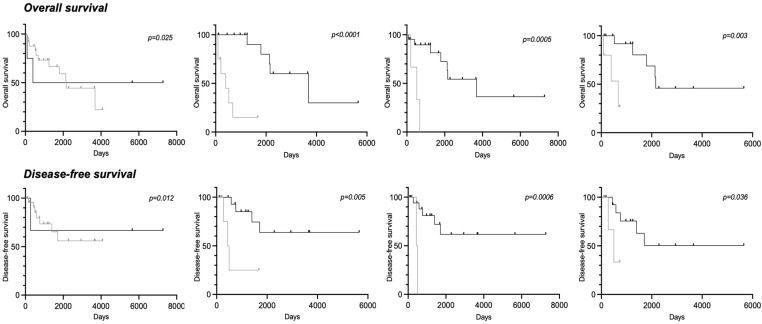
Correlations between RANKL/CD163/CD68/CD4^+^ expression and patient outcomes. Kaplan–Meier curves showing the association between the expression of the biomarkers with overall survival and disease-free survival.

**Table 1 cancers-15-01004-t001:** Patient characteristics. n, number of patients; S.D., standard deviation; F, female; M, male.

Patient Characteristics	
Age, *n* (years) ± S.D. (min–max)	47.8 ± 19.9 (17.3–83.9)
Gender: F (%)/M (%)	22 (44.0)/28 (56.0)
Grade of malignity, *n* (%)	
High-grade	40 (80.0)
Intermediate grade	7 (14.0)
Unknown	3 (6.0)
Histological subtype, *n* (%)	
Chondroblastic	23 (46.0)
Osteoblastic	16 (32.0)
Fibroblastic	9 (18.0)
Undifferentiated	2 (4.0)
Response to the neoadjuvant chemotherapy, *n* (%)	
Poor	20 (40.0)
Good	6 (12.0)
Unknown	24 (48.0)
Progression of the disease, *n* (%)	
Local recurrence	6 (12.0)
Metastases	7 (14.0)
Overall survival, *n* (days) ± S.D.	1600 ± 1737
Disease-free survival, *n* (days) ± S.D.	1469 ± 1745

**Table 2 cancers-15-01004-t002:** Biomarker staining results and correlations. n, number of patients; S.D., standard deviation; ^a^ Spearman’s rank correlation coefficient; ^b^, significance level.

Biomarker Staining Results							
Antibody	Nb Tested	Mean Positive Cells, *n* ± S.D. (Min–Max)		Nb ≥ 50% Positive Cells (%)		Nb ≥ 10% Positive Cells (%)	Nb ≥ 1% Positive Cells (%)
RANK	27	62.9 ± 33.7 (0–90)		-		22 (81.5%)	-
RANKL	35	70.0 ± 29.5 (10–100)		-		30 (85.7%)	-
CD146	35	-		20 (71.4%)		-	-
CD163	28	37.7 ± 21.6 (1–70)		9 (32.1%)		-	-
CD68	28	21.4 ± 17.9 (1–70)		3 (10.7%)		-	-
CD4^+^	24	13.7 ± 19.8 (0–50)		-		8 (33.3%)	-
CD8^+^	25	8.9 ± 15.2 (1–50)		-		-	7 (28%)
PD-1	27	0.1 ± 0.4 (0–1)		-		-	4 (14.8%)
PD-L1	24	2.1 ± 10.2 (0–50)		-		-	1 (4.2%)
**Correlations between Biomarkers**							
	RANK	RANKL	CD163	CD68	CD4^+^	CD8^+^	PD-1
RANKL	0.1232 ^a^ 0.5404 ^b^						
CD163	0.0526 ^a^ 0.8115 ^b^	−0.3261 ^a^ 0.0904 ^b^					
CD68	−0.0110 ^a^ 0.9582 ^b^	−0.1714 ^a^ 0.3833 ^b^	0.6956 ^a^ **<0.0001** ^b^				
CD4^+^	0.4718 ^a^ **0.0308** ^b^	−0.3373 ^a^ 0.1069 ^b^	0.5011 ^a^ **0.0149** ^b^	0.4607 ^a^ **0.0269** ^b^			
CD8^+^	0.1377 ^a^ 0.5627 ^b^	−0.2687 ^a^ 0.1940 ^b^	0.6715 ^a^ **0.0003** ^b^	0.3619 ^a^ 0.0897 ^b^	0.2971 ^a^ 0.1586 ^b^		
PD-1	0.1300 ^a^ 0.5545 ^b^	−0.1641 ^a^ 0.4134 ^b^	0.3576 ^a^ 0.0793 ^b^	0.3207 ^a^ 0.1265 ^b^	0.5781 ^a^ **0.0031** ^b^	0.2872 ^a^ 0.1736 ^b^	
PD-L1	0.0569 ^a^ 0.8064 ^b^	−0.3234 ^a^ 0.1232 ^b^	0.1939 ^a^ 0.3753 ^b^	0.0816 ^a^ 0.7113 ^b^	0.2927 ^a^ 0.1652 ^b^	0.3788 ^a^ 0.0747 ^b^	−0.0932 ^a^ 0.6647 ^b^

**Table 3 cancers-15-01004-t003:** Univariate analysis of overall survival and disease-free survival. HR, hazard ratio; CI, confidence interval; PR, poor responder; GR, good responder.

	Overall Survival		Disease-Free Survival	
	HR 95% CI	*p*	HR 95% CI	*p*
Age ≥ 50 y	1.44 [0.72–2.88]	0.27	0.67 [0.33–1.35]	0.23
Sex, female vs. male	1.37 [0.67–2.82]	0.35	1.48 [0.70–3.12]	0.25
Histological subtype				
Chondro vs. osteo	0.88 [0.38–2.02]	0.74	0.82 [0.35–1.92]	0.61
Chondro vs. other	0.89 [0.45–1.75]	0.72	0.61 [0.28–1.34]	0.23
Histopathologic grade				
High vs. intermediate	0.72 [0.32–1.61]	0.42	1.44 [0.66–3.15]	
PR vs. GR	2.78 [1.13–6.83]	**0.02**	3.02 [1.19–7.61]	**0.01**
Progression of the disease	1.45 [0.62–3.37]	0.31	1.66 [0.66–4.21]	0.19
RANK ≥ 10%	1.32 [0.49–3.56]	0.60	2.88 [1.24–6.68]	**0.03**
RANKL ≥ 10%	2.11 [0.92–4.85]	0.08	2.88 [1.24–6.68]	**0.03**
CD146 ≥ 50%	1.27 [0.62–2.60]	0.46	1.47 [0.71–3.24]	0.28
CD163 ≥ 50%	3.62 [1.06–12.36]	**0.0006**	3.24 [0.91–27.55]	0.003
CD68 ≥ 50%	3.03 [0.44–20.84]	0.04	3.21 [0.44–23.28]	0.04
CD4^+^ ≥ 10%	4.09 [0.78–21.60]	0.002	4.25 [0.77–23.50]	0.001
CD8^+^ ≥ 1%	1.84 [0.55–6.19]	0.21	1.55 [0.44–5.52]	0.66
PD-1 ≥ 1%	3.02 [0.44–20.72]	0.05	2.65 [0.43–16.27]	0.09
PD-L1 ≥ 1%	3.71 [0.09–15.3]	0.16	3.71 [0.09–15.3]	0.16

## Data Availability

The datasets used and analyzed during the current study are available from the corresponding author on reasonable request.

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
