# Peer review of "Characterization of the Tumor Microenvironment in Jaw Osteosarcomas, towards Prognostic Markers and New Therapeutic Targets"

_cancers, 2023, doi:10.3390/cancers15041004_

Round 1

Reviewer 1 Report

The authors show en efficient organisation of a clinical research consortium for the anatomic-pathological analyses of rare jaw osteosaromas (JO) and the tumor environment. They concluded that the immunohistochemical staining of CD163+ TAMs represent markers of poor prognosis in JO. In long-bone osteosarcomas (LBO) the opposite was found?

They explain these contradictory results as consequence of differences in biopsy methods in which the direct tumor environment (that contains most of the TAMs) is more or less sampled, see lines 330 - 337.  Can the authors suggest a biopsy protocol to avoid this? 

If CD163+ cells can differentiate into osteoclasts and or macrophages, can the authors suggest a better immuno-staining of specific 'type 2' macrophages or TAMs?

Reviewer 2 Report

The authors Bertin and colleagues performed an immunohistochemical study involving 50 patients with jaw osteosarcoma, either primitive or secondary to irradiation or to another sarcoma, with the aim to investigate the tumor microenvironment with specific focus on bone resorption, vascular and immune infiltrates. The authors observed a correlation between CD163 staining and worse OS, as well as high RANK/RANKL levels in patients with worse DFS.

The topic is of great interest since it focuses on the role of TME in disease progression and offers an interesting insight on its markers potential role as prognostic tools.

The manuscript is well written and organised, tables and figures are clear and conclusions are supported by data.

However, the following suggestions should be taken into account in order to improve the paper:

1) in Table 1 as well as in Results section the authors report the percentage/numbers of patients responding to neoadj CT, however it would be relevant to mention also which treatments the patients received. Indeed, it is only generically mentioned that standard treatment for JO is methotrexate+cisplatin/ifo/eto/doxo, however treatment schedule (doses, type of combination, cycles) could affect the TME and is turn the level of immune infiltrate, as well as the patient outcome itself. Same thing for adj CT, if administered to any patient. If the group is not homogenous in terms of pharmacological treatment received, it would be useful to stratify the patients for this criteria, or ate least mention the regimens used.

2) in Figure 2, the y axis is missing a label. Please add 

3) in Discussion section, expanding the paragraph on the role of bone remodelling biomarkers and vascularization in bone sarcomas would be and added value to the discussion.

In this regard, in a recent work Palmerini and colleagues highlighted that elevated baseline s-CTX levels correlated with an higher risk of progression in GCTB disease. 

Moreover, the role of angiogenesis in tumorigenesis is not only related to dissemination propensity. Indeed, its role and correlation with RANK/RANKL axis has been investigated in previous years in several works, in which the involvement of VEGFR  has been described in supporting RANKL-induced osteoclastogenesis. In this regard, recent evidence exists about the greater efficacy of combination of the anti-VEGFR lenvatinib and the anti-RANKL denosumab in the treatment of GCTB primary cultures, compared to denosumab alone. Moreover, the potential efficacy of TKI treatment in bone sarcomas is also emerging from ongoing studies on bone sarcomas (see clinical trial NCT04784247). 

Please discuss these findings adding the following relevant references:

Palmerini E et al. Bone Turnover Marker (BTM) Changes after Denosumab in Giant Cell Tumors of Bone (GCTB): A Phase II Trial Correlative Study. Cancers. 2022; 14(12):2863. https://doi.org/10.3390/cancers14122863 

- Gaspar, N. et al. "Lenvatinib with etoposide plus ifosfamide in patients with refractory or relapsed osteosarcoma (ITCC-050): A multicentre, open-label, multicohort, phase 1/2 study." Lancet Oncol. 2021, 22, 1312–1321

De Vita, A. et al. A Rationale for the Activity of Bone Target Therapy and Tyrosine Kinase Inhibitor Combination in Giant Cell Tumor of Bone and Desmoplastic Fibroma: Translational Evidences. Biomedicines 202210, 372. 

Round 2

Reviewer 2 Report

The authors assessed all the points raised by the reviewer, providing detailed informations about the treatments that patients received and explaining the reasons for missing informations.

The manuscript overall quality improved greatly and is now suitable for publication.